# Learn to Memorize: Optimizing LLM-based Agents with Adaptive Memory Framework

## Abstract

LLM-based agents have been extensively applied across various domains, where memory stands out as one of their most essential capabilities. Previous memory mechanisms of LLM-based agents are manually predefined by human experts, leading to higher labor costs and suboptimal performance. In addition, these methods overlook the memory cycle effect in interactive scenarios, which is critical to optimizing LLM-based agents for specific environments. To address these challenges, in this paper, we propose to optimize LLM-based agents with an adaptive and data-driven memory framework by modeling memory cycles. Specifically, we design an MoE gate function to facilitate memory retrieval, propose a learnable aggregation process to improve memory utilization, and develop task-specific reflection to adapt memory storage. Our memory framework empowers LLM-based agents to learn how to memorize information effectively in specific environments, with both off-policy and on-policy optimization. In order to evaluate the effectiveness of our proposed methods, we conduct comprehensive experiments across multiple aspects. To benefit the research community, we release our project at `https://anonymous.4open.science/r/learn_to_memorize`.

## 1 Introduction

Large language model (LLM) based agents have been widely applied in various fields (Wang et al., 2024; Xi et al., 2025; Guo et al., 2024), such as finance (Ding et al., 2024), recommender systems (Zhang et al., 2025a), and personal assistants (Li et al., 2024). During the interaction with environments, agents are supposed to perceive and memorize observations to support subsequent decision-making processes. These memories are crucial for maintaining the consistency of contextual interactions, and providing necessary information to facilitate reasoning under the current environment (Zhang et al., 2024a). Previous studies have proposed various methods to construct memory of LLM-based agents. These methods primarily rely on retrieval-augmented generation (RAG) (Gao et al., 2023) to acquire relevant information about the current states for in-context learning (Dong et al., 2022; Zhong et al., 2024). Recent approaches also explore transforming observations into modifications of model parameters to implement memories (Yang et al., 2024).

However, there are two significant limitations in previous studies. First of all, most of these methods are manually predefined by human experts, lacking a data-driven optimization process. For instance, Park et al. (2023) introduce a retrieval function by considering different aspects of memories. However, it assigns weights to these aspects manually. Similarly, MemoryBank (Zhong et al., 2024) summarizes critical information from observations before storage, yet it relies on fixed and intuitive prompts. In such cases, human experts need to try numerous parameters for better performance, resulting in increased labor costs and suboptimal performance, as demonstrated in **Figure 1(a)**.

Moreover, previous studies have largely overlooked the *memory cycle effect*, as illustrated in **Figure 1(b)**, which highlights a significant difference between vanilla LLMs and LLM-based agents. For vanilla LLMs, due to the lack of feedback from environments, they fail to establish a cycle between memory storage and utilization. In contrast, LLM-based agents store observations from environments as memories, in order to support the subsequent reasoning to take actions. These actions will further influence the states of environments, resulting in new feedback as observations that will be stored in the next cycle. From this perspective, the policies of memory storage and utilization mutually influence each other during agent-environment interactions. Therefore, learning either of them in isolation may lead to suboptimal performance due to neglecting the memory cycle effect.

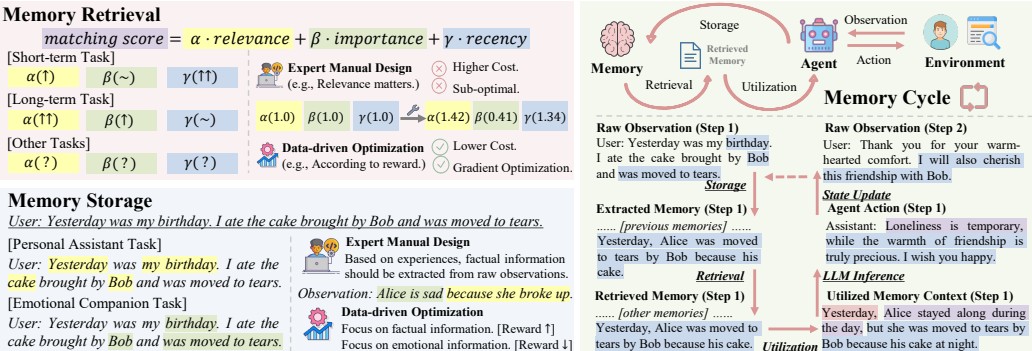

**(a) Manual vs. Data-driven Memory Design**    **(b) Memory Cycle Effect**

Figure 1: (a) In memory retrieval, the optimal weights for different aspects vary across different tasks. Similarly, in memory storage, the attention of information storage is task-dependent. However, manual model adaptation by human experts results in higher labor costs and suboptimal performance. (b) We demonstrate the memory cycle during interactions between agents and environments.

In this paper, we propose an adaptive memory framework that can be optimized with a data-driven approach. This framework formulates a memory cycle that consists of retrieval, utilization, and storage. Specifically, we design a Mixture-of-Expert (MoE) gate function across multiple aspects to implement adaptive combination for retrieval. We implement prompt optimization through task-specific reflection to adjust the extraction focus for storage. We propose a learnable aggregation process to better utilize retrieved memories, which is aligned by direct preference optimization (DPO) (Rafailov et al., 2023). In addition, to optimize our framework based on the training data, we propose off-policy and on-policy optimization strategies considering the memory cycle effect. Finally, we conduct extensive experiments to verify the effectiveness of our framework. To benefit the research community, we release our code on Github Repository[1].

Our primary contributions can be summarized as follows:
- We propose an adaptive and data-driven memory framework that empowers LLM-based agents to learn to memorize, with optimizable memory retrieval, utilization, and storage procedures.
- We formulate the memory cycle effect during agent-environment interactions, and propose off-policy and on-policy optimization strategies for our memory framework.
- We conduct comprehensive experiments to demonstrate the effectiveness of our framework in improving the performance of LLM-based agents when interacting with environments.

## 2 RELATED WORKS

### 2.1 REINFORCEMENT LEARNING BASED AGENTS

Reinforcement learning (RL) primarily studies how agents can optimize their actions within an environment to maximize cumulative rewards (Sutton et al., 1998). Unlike supervised learning, RL emphasizes learning by interacting with environments rather than relying on labeled data. Specifically, an RL-based agent makes decisions, receives feedback, and adjusts its strategy based on the results of its actions. The target is to establish a mapping from states to actions to maximize their rewards. Previous research has extensively explored optimizing RL-based agents, using methods such as Policy Gradient (Sutton et al., 1999), DQN (Mnih et al., 2013), Actor-Critic (Konda and Tsitsiklis, 1999), and DDPG (Lillicrap et al., 2015). These approaches typically construct cross-trial experiences through neural networks or tabular methods by exploration and exploitation.

### 2.2 LARGE LANGUAGE MODEL BASED AGENTS

With the rapid development of LLMs, building agents based on LLMs has emerged as a promising field of research (Wang et al., 2024; Xi et al., 2025; Guo et al., 2024). These LLM-based agents have recently found extensive applications in various domains, including finance (Ding et al., 2024), social simulation (Gao et al., 2024), and personal assistants (Li et al., 2024). For instance, Generative Agents (Park et al., 2023) aim to simulate human daily activities with LLM-based agents.

---

[1]https://anonymous.4open.science/r/learn_to_memorize

Although they commonly have different architectures (Xi et al., 2025), most of them incorporate reasoning (Huang et al., 2024a), memory (Zhang et al., 2024a), and action modules. Similar to RL-based agents, they also interact with environments by receiving observations, making decisions, and taking actions. However, due to their extensive pretraining, LLM-based agents have more prior world knowledge, which enhances their generalization capabilities.

### 2.3 Memory of Large Language Model Based Agents

For LLM-based agents, memory is one of the most critical capabilities for interacting with environments, as it maintains contextual consistency and supports inferences made by LLMs (Zhang et al., 2024a). Previous studies primarily employ in-context learning to implement memories (Park et al., 2023; Zhong et al., 2024). They commonly utilize RAG methods to retrieve relevant information about the current states and incorporate it into prompts (Gao et al., 2023). For example, Memory-Bank (Zhong et al., 2024) implements a hierarchical memory structure with textual summarization and forgetting mechanisms. MemTree (Rezazadeh et al., 2024) proposes a tree-structured memory framework that dynamically updates memory storage. However, they still require human experts to manually design for specific applications, and they overlook the memory cycle effect during interactions. These limitations result in increased labor costs and suboptimal performance.

## 3 Methods

### 3.1 Preliminary: Memory Cycle Effect

Before proposing our adaptive memory framework, we explicitly formulate the memory cycle, as illustrated in **Figure 2(a)**. We model the continuous interactions between agents and environments as a Markov Decision Process (MDP) (Sutton et al., 1998). Specifically, we denote the state transition distribution of an environment as $p_{\text{env}}(\cdot|s^t, a^t)$, where $s^t$ and $a^t$ represent the state and action at step $t$. We employ the reward function $r(s^t, a^t)$ to reflect the achievement of tasks. We denote the agent's policy as $\pi_{\text{agent}}(\cdot|s^t, \theta)$, where the next action is determined by the current state $s^t$ with parameter $\theta$. For LLM-based agents, their policies are typically implemented with memory contexts to construct prompts for LLMs. During the interaction process, at each step $t$, the agent perceives the current state $s^t$, and selects an action by $a^t \sim \pi_{\text{agent}}(\cdot|s^t, \theta)$. Then, the state is updated by $s^{t+1} \sim p_{\text{env}}(\cdot|s^t, a^t)$, obtaining the reward $r(s^t, a^t)$. The objective is to maximize cumulative reward.

The memory cycle effect in agent-environment interactions can be further formulated into a framework with three consecutive procedures as demonstrated in **Figure 2(b)**, including memory storage $S(\theta_s; \cdot)$, retrieval $R(\theta_r; \cdot)$, and utilization $U(\theta_u; \cdot)$. Here, we use $\theta = \{\theta_s, \theta_r, \theta_u\}$ to emphasize their model parameters. First, the agent observes the current state $s^t$ and stores it into the storage $M^t$, where $M^t = S(\theta_s; M^{t-1}, s^t)$. Then, the agent retrieves a ranked subset of the current storage by $M^t_{\text{rank}} = R(\theta_r; s^t, M^t)$ based on the current state $s^t$. After that, the agent integrates this memory subset into a prompt through the utilization procedure by $p^t = U(\theta_u; M^t_{\text{rank}}, s^t)$. Finally, the agent determines the next action by LLM with $a^t = \text{LLM}(p^t)$, and updates the state $s^{t+1} \sim p_{\text{env}}(\cdot|s^t, a^t)$ for the next cycle. In these cycles, the memory storage, retrieval, and utilization procedures are not isolated, but influence each other. Therefore, the optimization of these three procedures should be performed jointly. Our adaptive memory framework is proposed based on this framework of the memory cycle effect, which can be optimized in a data-driven manner. An overview of our framework is demonstrated in **Figure 2(c)**, and we present the details in the rest of this section.

### 3.2 Memory Retrieval Procedure

Due to the large collection of memories, agents should retrieve a subset of memories before integrating them into prompts. Previous studies typically calculate matching scores $f(s^t, m_i)$ between the current state $s^t$ and memories $m_i \in M^t$, and select the top-$k$ memories. These matching scores are often associated with metrics, such as semantic similarity, time recency, and so on. For instance, Generative Agents (Park et al., 2023) calculate the matching scores by $f(s^t, m_i) = \alpha_{\text{rel}} \cdot d_{\text{rel}}(s^t, m_i) + \alpha_{\text{imp}} \cdot d_{\text{imp}}(s^t, m_i) + \alpha_{\text{rec}} \cdot d_{\text{rec}}(s^t, m_i)$, where $d_{\text{rel}}(\cdot), d_{\text{imp}}(\cdot), d_{\text{rec}}(\cdot)$ are metric functions and $\alpha_{\text{rel}}, \alpha_{\text{imp}}, \alpha_{\text{rec}}$ are weights on semantic relevance, memory importance, and time recency. However, these weights are fixed and manually determined by human experts, instead of learning from interactions with the environment. Besides, in different applications, the

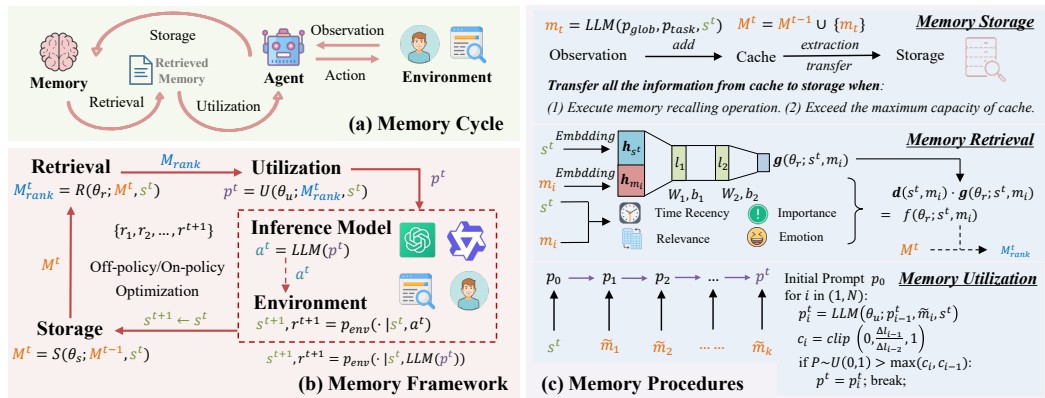

Figure 2: Overview of the memory cycle effect and adaptive memory framework.

significance of metric functions can also be various, and the sensitivities of states and memories on different metrics are often not the same.

To solve these challenges, we propose an optimizable retrieval procedure. We define a vector-valued function $\mathbf{d}(s^t, m_i) = [d_1(s^t, m_i), d_2(s^t, m_i), ..., d_n(s^t, m_i)]$, where $d_1(\cdot), d_2(\cdot), ..., d_n(\cdot)$ are metric functions. Then, we design a parameterized MoE gate function to activate different metrics as $\mathbf{g}(\theta_r; s^t, m_i) = [g_1(\theta_r; s^t, m_i), g_2(\theta_r; s^t, m_i), ..., g_n(\theta_r; s^t, m_i)]$, where $\theta_r$ represents optimizable parameters. This gate function can adaptively adjust weights on metric functions for different states and memories based on the training data. After that, we calculate the matching scores by $f(\theta_r; s^t, m_i) = \mathbf{g}(\theta_r; s^t, m_i) \cdot \mathbf{d}(s^t, m_i)^T$. Finally, all the memories $m_i \in M^t$ are ranked according to their matching scores, resulting in a ranked memory list $M_{\text{rank}}^t = [\tilde{m}_1^t, \tilde{m}_2^t, ..., \tilde{m}_t^t]$.

In addition, we extend metric functions to expand the learning space for covering more potential retrieval policies. We incorporate emotional relevance by pre-training a scoring function to extract emotions from memories. We further extend linear time recency using Taylor's Formula to get $d_{\text{rec}}^p(s^t, m_i) = ||\frac{\Delta(s^t, m_i)}{t}||_p$ on various $p$-norms, where $\Delta(s^t, m_i)$ is the time gap between $s^t$ and $m_i$. The full details are provided in **Appendix A**. Besides, we implement $\mathbf{g}(\theta_r; s^t, m_i)$ with

$$\mathbf{g}(\theta_r; s^t, m_i) = \text{softmax}\left(W_2 \cdot \sigma(W_1 \cdot [\mathbf{h}_{s^t}; \mathbf{h}_{m_i}]^T + b_1) + b_2\right),$$

where $\theta_r = \{W_1, W_2, b_1, b_2\}$ are optimizable parameters, and $\mathbf{h}_{s^t}, \mathbf{h}_{m_i}$ are embeddings of $s^t, m_i$.

### 3.3 MEMORY UTILIZATION PROCEDURE

After obtaining the retrieval result $M_{\text{rank}}^t$, it is necessary to transform it into a memory context to serve as part of the prompt $p^t$. Most studies directly concatenate their top-$k$ memories. However, this approach solely focuses on state-memory relations but overlooks memory-memory relations. It leads to the recurrence of similar memories within the same context. To solve this problem, we design a learnable memory augmentation process that can be optimized using training datasets.

We iteratively integrate the memories from $M_{\text{rank}}^t$ into the memory context. Starting with the initial memory context $p_0^t$, we obtain $p_i^t = \text{LLM}(\theta_u; p_{i-1}^t, \tilde{m}_i^t, s^t)$ for $i \geq 1$ until the end of process, where $\theta_u$ represents the optimizable parameters in LLMs. To prevent excessive merging steps, we calculate the word increase rate from $p_{i-1}^t$ to $p_i^t$ as $\Delta l_i^t$, and approximate the information gain as $c_i = \text{clip}(\frac{\Delta l_i}{\Delta l_{i-1}}, 0, 1)$. Then, we sample the stop signal with $z_i \sim B\left(1 - \max(c_i, c_{i-1})\right)$ to allow one exemption, where $B(\cdot)$ denotes a Bernoulli distribution. Finally, we incorporate the last memory context into the template to get the prompt $p^t$. However, common LLMs may exhibit suboptimal performance for specific applications. To address this issue, we adjust the parameters $\theta_u$ of LLMs to align with training datasets through SFT and DPO, as described in **Section 4**.

### 3.4 MEMORY STORAGE PROCEDURE

When an agent perceives a new state during interactions, it typically extracts critical parts from complete observations before storing them. For instance, an agent designed for personal assistance should concentrate on factual daily information from observations (Zhang et al., 2024b), while an emotional companion agent should prioritize sentiments (Zhong et al., 2024). This extraction pro-

---

**Algorithm 1:** Algorithm of on-policy optimization.

---

**Input:** The number of total epochs $L$, the trajectory size $n$, the learning rates $\alpha_r, \alpha_u^s, \alpha_u^d$, and the initial parameters $\theta_s^0, \theta_r^0, \theta_u^0$.

**Output:** The optimized parameters $\theta_s^*, \theta_r^*, \theta_u^*$.

1 **for** $l \leftarrow 1$ **to** $L$ **do**

2     Sample trajectories $T_1, T_2, ..., T_n$ by interacting with the training environment.

3     $\theta_r^l \leftarrow \theta_r^{l-1} - \alpha_r \cdot \nabla \frac{1}{n} \sum_{i=1}^{n} \frac{1}{t_i} \sum_{j=1}^{t_i} w_{i,j} \ln \frac{\sigma\left[f(\theta_r^{l-1}; s_i^{t_i}, \tilde{m}_{i,t_i-j+1}^{t_i}) - f(\theta_r^{l-1}; s_i^{t_i}, \tilde{m}_{i,j}^{t_i})\right]}{\sigma\left[f(\theta_r^{l-1}; s_i^{t_i}, \tilde{m}_{i,j}^{t_i}) - f(\theta_r^{l-1}; s_i^{t_i}, \tilde{m}_{i,t_i-j+1}^{t_i})\right]}$.

4     $\theta_u^l \leftarrow \theta_u^{l-1} - \alpha_u^s \cdot \nabla \frac{1}{n} \sum_{i=1}^{n} \text{CELoss}(\theta_u^{l-1}; \tilde{p}_{t_i}^{t_i} | p_{t_i-1}^{t_i}, \tilde{m}_{t_i}^{t_i}, s_i^{t_i})$.

5     $\theta_u^l \leftarrow \theta_u^l - \alpha_u^d \cdot \nabla \frac{1}{n} \sum_{i=1}^{n} \ln \sigma \left[ \beta \ln \frac{P(\theta_u^l; \hat{p}_{t_i}^{t_i} | p_{t_i-1}^{t_i}, \tilde{m}_{i,t_i}^{t_i}, s_i^{t_i})}{P(\theta_u^{l-1}; \hat{p}_{t_i}^{t_i} | p_{t_i-1}^{t_i}, \tilde{m}_{i,t_i}^{t_i}, s_i^{t_i})} - \beta \ln \frac{P(\theta_u^l; p_{t_i}^{t_i} | p_{t_i-1}^{t_i}, \tilde{m}_{i,t_i}^{t_i}, s_i^{t_i})}{P(\theta_u^{l-1}; p_{t_i}^{t_i} | p_{t_i-1}^{t_i}, \tilde{m}_{i,t_i}^{t_i}, s_i^{t_i})} \right]$.

6     $\theta_u^l \leftarrow \theta_u^{l-1} \bigcup_{i=1}^{n} \text{LLM}(\{s_i^{t_i}, m_i^{t_i}\} \in T_i^{\text{pos}}) \cup \text{LLM}(\{s_i^{t_i}, m_i^{t_i}\} \in T_i^{\text{neg}})$.

7 **end**

8 Obtain optimized parameters $\theta_s^* = \theta_s^L, \theta_s^r* = \theta_r^L, \theta_u^* = \theta_u^L$.

9 **return** $\theta_s^*, \theta_r^*, \theta_u^*$.

---

cess can be implemented by LLM to highlight critical aspects in instructions. However, different applications inherently emphasize distinct aspects.

To solve this problem, we design an extraction process based on task-specific reflection (Shinn et al., 2023). Specifically, we structure an instruction with a general part $p_{\text{glob}}$ and a task-specific part $p_{\text{task}}$. Then, we consider $p_{\text{task}}$ as the learnable parameter $\theta_s$, and optimize it based on successful and unsuccessful trajectories from training datasets, as discussed in the next section. For each interaction, we transform an observation into a memory unit with $m_t = \text{LLM}(p_{\text{glob}}, p_{\text{task}}, s^t)$, and update the storage with $M^t = M^{t-1} \cup \{m_t\}$. To balance the extraction load, we set a cache to temporarily hold observations, and transfer them into storage when recalling memories or reaching the capacity.

## 4 OPTIMIZATION STRATEGIES

### 4.1 OVERVIEW: OPTIMIZATION OF LLM-BASED AGENTS

Unlike LLM optimization, which is based on a static corpus, optimizing LLM-based agents relies on interactions with dynamic environments. Therefore, we propose two strategies to optimize our memory framework. The first strategy is off-policy optimization, which samples trajectories $\mathcal{D}$ from training environments using the reference policy $\pi_{\text{agent}}(\cdot | s^t, \theta^{\text{ref}})$, and optimizes another policy $\pi_{\text{agent}}(\cdot | s^t, \theta)$ with the loss function $\mathcal{L}(\cdot)$. It supports offline training and the reuse of previous trajectories, making it more flexible and efficient. However, it encounters the issue of distribution shift between the sampling policy $\pi_{\text{agent}}(\cdot | s^t, \theta^{\text{ref}})$ and the optimized policy $\pi_{\text{agent}}(\cdot | s^t, \theta)$. Another strategy is on-policy optimization, which consistently employs the optimized policy $\pi_{\text{agent}}(\cdot | s^t, \theta)$ to sample training trajectories for ongoing optimization. This approach requires online learning to keep alignment between the sampling and optimized policies, thereby alleviating distribution shifts.

### 4.2 OFF-POLICY OPTIMIZATION

**Memory Retrieval Optimization.** We propose a contrastive learning approach to optimize parameters $\theta_r$ of the MoE gate function $\mathbf{g}(\theta_r; s^t, m_i)$ in the retrieval procedure. First of all, we filter out all the successful interactions $\mathcal{D}_s$ whose final rewards exceed the threshold $\beta_r$ (*e.g.,* answer correctly for QAs). Then, we focus on the ranking result $M_{\text{rank}}^t = [\tilde{m}_1^t, \tilde{m}_2^t, ..., \tilde{m}_t^t]$ from their memory retrieval procedures. We pair all the elements $\tilde{m}_i^t \in M_{\text{rank}}^t$ in reverse order as $x_i = (\tilde{m}_i^t, \tilde{m}_{t-i+1}^t)$, where $1 \leq i \leq t$. Then, we assign a weight $w_i = \frac{-\text{sign}(v_i)}{\sum_{j=1}^{t} \gamma^{v_i}} \cdot \gamma^{v_i}$ with $v_i = t - 1 - |t - 2j + 1|$, which allocates higher contrastive confidence to the pairs with larger ranking differences, thereby reducing the ranking noise. Finally, we define our loss function as

$$\mathcal{L}(\theta_r; \mathcal{D}_s) = \frac{1}{|\mathcal{D}_s|} \sum_{s^t, M_{\text{rank}}^t \in \mathcal{D}_s} \frac{1}{t} \sum_{i=1}^{t} w_i \cdot \ln \frac{\sigma\left[f(\theta_r; s^t, \tilde{m}_{t-i+1}^t) - f(\theta_r; s^t, \tilde{m}_i^t)\right]}{\sigma\left[f(\theta_r; s^t, \tilde{m}_i^t) - f(\theta_r; s^t, \tilde{m}_{t-i+1}^t)\right]},$$

and optimize the parameters with $\theta_r^* = \arg\min_{\theta_r} \mathcal{L}(\theta_r; \mathcal{D})$ by gradient descent.

**Memory Utilization Optimization.** We employ SFT and DPO to optimize the parameters $\theta_u$ of LLMs for the aggregation process in memory utilization. First of all, we choose the final interactions of training trajectories and denote them as $\mathcal{D}_l$. Then, we focus on their memory utilization procedures. Specifically, we optimize $\theta_u$ in $p_t^t = \text{LLM}(\theta_u; p_{t-1}^t, \tilde{m}_t^t, s^t)$ by collecting the outputs from expert models in $\tilde{p}_t^t = E(p_{t-1}^t, \tilde{m}_t^t, s^t)$, where the expert model $E(\cdot)$ can be implemented by domain-specific or more advanced LLMs. Finally, the SFT loss function can be expressed as

$$\mathcal{L}^{\text{SFT}}(\theta_u; \mathcal{D}_l) = \frac{1}{|\mathcal{D}_l|} \sum_{p_{t-1}^t, \tilde{m}_t^t, s^t \in \mathcal{D}_l} \text{CELoss}(\theta_u; \tilde{p}_t^t | p_{t-1}^t, \tilde{m}_t^t, s^t),$$

where $\text{CELoss}(\cdot)$ is the cross-entropy loss function. Then, we have $\theta_u^{\text{SFT}} = \arg\min_{\theta_u} \mathcal{L}^{\text{SFT}}(\theta_u; \mathcal{D}_l)$. We further refine the expert model using DPO to better align with the expert model. Specifically, we consider $\text{LLM}(\theta_u^{\text{SFT}}; \cdot)$ as the reference model, and re-generate utilization results with $\hat{p}_t^t = \text{LLM}(\theta_u; \hat{p}_{t-1}^t, \tilde{m}_t^t, s^t)$. Then, we establish the DPO loss function

$$\mathcal{L}^{\text{DPO}}(\theta_u; \mathcal{D}_l) = \frac{1}{|\mathcal{D}_l|} \sum_{p_{t-1}^t, \tilde{m}_t^t, s^t \in \mathcal{D}_l} \ln \sigma \left[ \beta \ln \frac{P(\theta_u; \hat{p}_t^t | p_{t-1}^t, \tilde{m}_t^t, s^t)}{P(\theta_u^{SFT}; \hat{p}_t^t | p_{t-1}^t, \tilde{m}_t^t, s^t)} - \beta \ln \frac{P(\theta_u; p_t^t | p_{t-1}^t, \tilde{m}_t^t, s^t)}{P(\theta_u^{SFT}; p_t^t | p_{t-1}^t, \tilde{m}_t^t, s^t)} \right],$$

where $\beta$ is a parameter to control the deviation from the original parameter $\theta_u^{\text{SFT}}$, and $P(\cdot)$ is the output probability distribution of the LLMs given certain parameters. Finally, we obtain the optimal parameters by $\theta_u^* = \arg\min_{\theta_u} \mathcal{L}^{\text{DPO}}(\theta_u; \mathcal{D}_l)$, where $\theta_u$ is initialized as $\theta_u^{\text{SFT}}$.

**Memory Storage Optimization.** To optimize the task-specific instruction for memory extraction, we optimize $\theta_s$ by self-reflection. Specifically, we divide all the interactions into two groups based on their rewards. The interactions with rewards above the threshold $\beta_s$ are placed in the positive group $\mathcal{D}_{\text{pos}}$, while interactions with rewards below the threshold are assigned to the negative group $\mathcal{D}_{\text{neg}}$. For interactions in the positive group, we utilize LLMs to reflect and summarize their successful experiences. Similarly, the failure experiences can also be reflected and summarized by LLMs automatically. After that, we iteratively update the task-specific prompt with $p_{\text{task}} \leftarrow p_{\text{task}} \cup \text{LLM}(\{s^t, m^t\} \in \mathcal{D}_{\text{pos}}) \cup \text{LLM}(\{s^t, m^t\} \in \mathcal{D}_{\text{neg}})$, where we have $\theta_s^* = p_{\text{task}}^*$.

### 4.3 On-policy Optimization

Although off-policy optimization supports offline training, it is often hindered by distribution shifts between the sampling policy and the optimized policy, leading to suboptimal performance in memory cycles. To alleviate this problem, we extend our optimization strategy to on-policy optimization, as described in **Algorithm 1**. Building upon the model parameters after off-policy optimization, we further conduct the on-policy optimization with online learning. Specifically, during each epoch, we sample $n$ trajectories by interacting with the training environment based on the current model parameters. Then, we utilize the training procedures above with single-step optimization to update model parameters. Finally, we obtain the optimal model parameters from the last epoch.

## 5 Experiments

### 5.1 Experimental Settings

Our experiments are conducted on three datasets with various difficulty levels, including HotpotQA-hard, HotpotQA-medium, and HotpotQA-easy (Yang et al., 2018). Additionally, we also carry out experiments on MemDaily (Zhang et al., 2024b), and put the details in **Appendix B** due to the page limitation. To fulfill interactive scenarios between the agent and the environment, we adopt *fullwiki* mode in HotpotQA by implementing a simulator to create a dynamic environment, which presents greater challenges than *distractor* mode with static references. For the LLM-based agents, we employ the ReAct (Yao et al., 2023) reasoning structure along with textual memory contexts (Zhang et al., 2024a). Our memory framework is compared against several baselines of memory models implemented by MemEngine (Zhang et al., 2025b) as follows:

- **FUMemory** (Full Memory): Directly concatenate all the observations into a memory context.
- **LTMemory** (Long-term Memory): Retrieve most relevant observations by semantic similarities.
- **STMemory** (Short-term Memory): Keep the latest observations to combine as a memory context.
- **GAMemory** ((Park et al., 2023)): Memory with self-reflection and weighted retrieval.
- **MBMemory** ((Zhong et al., 2024)): Hierarchical memory with summarization and forgetting.

Table 1: Overall performance across different baselines and inference models on various datasets. Bold values represent the best results, while underlined values represent the second-best results.

| HotpotQA-Hard | | | | | |
| --- | --- | --- | --- | --- | --- |
| **Inference** | **ActOnly** | **CoTOnly** | **FUMemory** | **LTMemory** | **STMemory** | **GAMemory** |
| GPT-4o-mini | 0.2832 | 0.3274 | 0.3451 | 0.3274 | 0.3540 | 0.3186 |
| Qwen-2.5 | 0.1504 | 0.2389 | 0.2920 | 0.2212 | 0.1504 | 0.2124 |
| Llama-3.1 | 0.1770 | 0.2566 | 0.1239 | 0.0619 | 0.0177 | 0.0354 |
| **Inference** | **MBMemory** | **SCMemory** | **MTMemory** | **Ours-def** | **Ours-off** | **Ours-on** |
| GPT-4o-mini | 0.3009 | 0.3363 | **0.3628** | 0.3274 | 0.3186 | 0.3274 |
| Qwen-2.5 | 0.2301 | 0.1416 | 0.2566 | 0.2832 | 0.2832 | **0.3186** |
| Llama-3.1 | 0.1062 | 0.0619 | 0.1504 | 0.2478 | 0.1416 | **0.2920** |
| HotpotQA-Medium | | | | | |
| **Inference** | **ActOnly** | **CoTOnly** | **FUMemory** | **LTMemory** | **STMemory** | **GAMemory** |
| GPT-4o-mini | 0.3303 | 0.4220 | **0.4862** | 0.4037 | 0.3945 | 0.3853 |
| Qwen-2.5 | 0.2202 | 0.2844 | 0.2844 | 0.2385 | 0.1651 | 0.1468 |
| Llama-3.1 | 0.1560 | 0.2294 | 0.1284 | 0.0642 | 0.0275 | 0.0642 |
| **Inference** | **MBMemory** | **SCMemory** | **MTMemory** | **Ours-def** | **Ours-off** | **Ours-on** |
| GPT-4o-mini | 0.3853 | 0.3486 | 0.3853 | 0.4220 | 0.4037 | 0.4404 |
| Qwen-2.5 | 0.2385 | 0.1009 | 0.2752 | 0.3119 | 0.3486 | **0.4037** |
| Llama-3.1 | 0.0642 | 0.0826 | 0.1743 | 0.2752 | 0.1468 | **0.3119** |
| HotpotQA-Easy | | | | | |
| **Inference** | **ActOnly** | **CoTOnly** | **FUMemory** | **LTMemory** | **STMemory** | **GAMemory** |
| GPT-4o-mini | 0.3738 | **0.4019** | 0.3645 | 0.3832 | 0.3832 | 0.3738 |
| Qwen-2.5 | 0.2991 | 0.3364 | 0.2710 | 0.2523 | 0.2056 | 0.1776 |
| Llama-3.1 | 0.2991 | **0.3271** | 0.1589 | 0.0654 | 0.0374 | 0.0935 |
| **Inference** | **MBMemory** | **SCMemory** | **MTMemory** | **Ours-def** | **Ours-off** | **Ours-on** |
| GPT-4o-mini | 0.3364 | 0.3645 | 0.3271 | 0.3832 | 0.3738 | 0.3738 |
| Qwen-2.5 | 0.2523 | 0.2056 | 0.3364 | 0.3925 | 0.3364 | **0.4112** |
| Llama-3.1 | 0.1028 | 0.0748 | 0.1495 | 0.2523 | 0.1682 | **0.3271** |

- **SCMemory** ((Wang et al., 2023)): Self-controlled memory with adaptive memory context length.
- **MTMemory** ((Rezazadeh et al., 2024)): Structured-based memory with node representation.

Besides, we implement two one-step baselines without memory for comparison as follows:
- **ActOnly**: Take actions based on current observations without memory or reasoning structure.
- **CoTOnly**: Reason on current observations by Chain-of-Thought (Wei et al., 2022) to take actions.

We represent our models as **Ours-def**, **Ours-off**, and **Ours-on**, corresponding to the non-optimized model, the off-policy optimized model, and the on-policy optimized model, respectively. Following the previous work (Yang et al., 2018), we calculate the accuracy of Exact Match (EM) between the ground-truth and predicted answer, serving as the final reward of each trajectory. Specifically, agents are required to answer a question in each independent trajectory. Within a maximum number of steps, they can either search for a keyword on Wikipedia in each step to get its full document, or submit a predicted answer to finish this trajectory. Due to the page limitation, we provide additional details regarding experimental settings in **Appendix D** to facilitate the reproduction.

## 5.2 Overall Performance

The results of major performances are present in **Table 1**. From the results, we find that our model with on-policy optimization outperforms other baselines in most cases, showing the effectiveness of our proposed framework. The results also reveal that our framework can still work with default parameters, showing a certain degree of multitask generalization, but its performance declines after off-policy optimization due to the distribution mismatch. Moreover, MTMemory and MBMemory also present relatively great performance, whereas the one-step baselines demonstrate weaknesses in more challenging tasks. Besides, it appears that some LLMs exhibit limited dependence on memory for easy-level questions, possibly because they have encountered the necessary references to these questions in their pre-training corpus. Finally, the performance of memory methods varies

Table 2: Results of ablation studies across different baselines, inference models and datasets. Bold values represent the best results, while underlined values represent the second-best results.

| HotpotQA-Hard | | | | | | | |
|---|---|---|---|---|---|---|---|
| Inference | Ours-def | Ours-R | Ours-U/sft | Ours-U/dpo | Ours-S | Ours-off | Ours-on |
| GPT-4o-mini | 0.3274 | **0.3451** | 0.3097 | 0.3186 | 0.2920 | 0.3186 | 0.3274 |
| Qwen-2.5 | 0.2832 | **0.3186** | 0.3009 | **0.3186** | 0.2832 | 0.2832 | **0.3186** |
| Llama-3.1 | 0.2478 | 0.2301 | 0.2478 | 0.1593 | 0.2566 | 0.1416 | **0.2920** |
| HotpotQA-Medium | | | | | | | |
| Inference | Ours-def | Ours-R | Ours-U/sft | Ours-U/dpo | Ours-S | Ours-off | Ours-on |
| GPT-4o-mini | 0.4220 | **0.4587** | 0.4220 | 0.4312 | 0.4495 | 0.4037 | 0.4404 |
| Qwen-2.5 | 0.3119 | 0.3303 | 0.3211 | 0.2661 | 0.3853 | 0.3486 | **0.4037** |
| Llama-3.1 | 0.2752 | 0.2385 | 0.2385 | 0.0917 | 0.2844 | 0.1468 | **0.3119** |
| HotpotQA-Easy | | | | | | | |
| Inference | Ours-def | Ours-R | Ours-U/sft | Ours-U/dpo | Ours-S | Ours-off | Ours-on |
| GPT-4o-mini | 0.3832 | **0.3925** | 0.3645 | 0.3551 | 0.3645 | 0.3738 | 0.3738 |
| Qwen-2.5 | 0.3925 | **0.4112** | 0.3271 | 0.3458 | 0.3645 | 0.3364 | **0.4112** |
| Llama-3.1 | 0.2523 | 0.2710 | 0.2243 | 0.1589 | 0.2617 | 0.1682 | **0.3271** |

across different inference models, potentially due to disparities in their in-context learning abilities to leverage memory contexts. Some methods show weak performance on open-source inference models, which may be attributed to the failure to organize effective memory contexts.

## 5.3 ABLATION STUDIES

In order to further study each procedure and optimization strategy within our framework, we conduct ablation experiments by independently optimizing retrieval, utilization (SFT/DPO), and storage procedures with off-policy optimization. We denote these ablation models as **Ours-R**, **Ours-U/sft**, **Ours-U/dpo**, and **Ours-S**, respectively. The results, presented in **Table 2**, indicate that on-policy optimization is crucial for improving the performance of our framework. Besides, optimizing individual memory procedures can also take effect, especially for the retrieval procedure. However, directly combining the parameters of memory procedures results in reduced performance. Intuitively, the memory procedures have mutual influence due to the memory cycle effect, but the off-policy samples fail to trace the optimized memory outcomes. Therefore, the optimal parameters of a certain procedure are based on the initial parameters of other procedures, leading to a policy mismatch.

## 5.4 EXTENSIVE STUDIES ON REASONING STEPS

To further study the effectiveness of memory methods inside trajectories, we calculate the average reasoning steps across different baselines under HotpotQA-hard and Qwen-2.5, and we present the results in **Figure 3**. The results indicate that our approach significantly reduces the average reasoning steps within trajectories. Specifically, the proportion of five-step reasoning in our model decreases, while the occurrence of two-step reasoning trajectories increases. Under identical conditions, achieving tasks with fewer reasoning steps suggests that agents can make more informed decisions utilizing memory, thereby finding answers more quickly and confidently. Meanwhile, we observe that models with lower overall performance generally require more inference steps. This might be due to their inability to solve the problem even after reaching the maximum steps.

## 5.5 EXTENSIVE STUDIES ON PRE-TRAINED METRIC FUNCTIONS

We further conduct experiments to verify the effectiveness of pre-trained metric functions in the memory retrieval procedure. Specifically, we calculate NDCG@5 for the ranked messages based on importance scoring and use MSE to evaluate emotion scoring across different dimensions. Additionally, we record the instruction failure rate (IFR) for prompting methods. Due to the page limitation, the detailed results are presented in **Appendix A.3**. The results indicate that our pre-trained metric functions show certain improvements over zero-shot and few-shot prompting methods in predicting importance scores and emotion decomposition. Moreover, we observe that some open-source models exhibit instruction failures during the scoring process, leading to instability in retrieval metrics.

Table 3: Results of time costs (seconds) across different baselines.

| Efficiency | ActOnly | CoTOnly | FUMemory | LTMemory | STMemory | GAMemory |
|---|---|---|---|---|---|---|
| Time/Step | 0.08 | 2.80 | 11.33 | 9.68 | 11.64 | 8.83 |
| Time/Trajectory | 0.08 | 2.80 | 43.05 | 32.91 | 54.73 | 39.72 |
| Efficiency | MBMemory | SCMemory | MTMemory | Ours-def | Ours-off | Ours-on |
| Time/Step | 8.38 | 7.14 | 107.34 | 14.98 | 13.03 | 11.74 |
| Time/Trajectory | 35.21 | 29.99 | 472.31 | 40.45 | 33.88 | 25.83 |


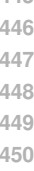
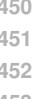

Figure 3: Results of average reasoning steps across different baselines.

## 5.6 INFLUENCE OF HYPER-PARAMETERS

We further explore the influence of some significant hyper-parameters in our framework, including SFT batch size, DPO batch size, and reflection batch size. Due to the page limitation, we include more details and the results in **Appendix C**. According to the results, we find that the best choice of SFT batch size is around 16, while the best DPO batch size is roughly 32. Additionally, we find that the accuracy is more sensitive to variations in DPO batch size, significantly diminishing when the values are particularly low. In contrast, the reflection batch size has a relatively minor impact on performance, with accuracy remaining similar when it ranges from 20 to 50 samples per reflection.

## 5.7 ANALYSIS OF EFFICIENCY

In addition to evaluating the effectiveness of memory mechanisms, we conduct experiments to assess their efficiency. Specifically, we calculate the average time cost of different baselines for each step and each trajectory. Our experiments are performed on a computing server with 8 NVIDIA A800-SXM-80G GPUs, and the results are presented in **Table 3**. According to the results, while our method shows a slight increase in time per step due to additional operations, the time per trajectory is significantly reduced because the total number of reasoning steps decreases. Additionally, we observe that FUMemory, LTMemory, and STMemory exhibit higher time consumption per step, possibly due to increased inference costs associated with longer memory contexts in prompts.

## 6 CONCLUSIONS AND LIMITATIONS

In conclusion, we propose an adaptive and data-driven memory framework to optimize LLM-based agents. We formulate the memory cycle with retrieval, utilization, and storage procedures. We develop an MoE gate function to enhance the memory retrieval procedure, a task-specific reflection process to refine the memory extraction, and a post-training stage to improve the memory utilization procedure. Additionally, we design both off-policy and on-policy optimization strategies based on the memory cycle effect. The extensive experiments have verified the effectiveness and efficiency of our methods. Despite the advancements achieved in our study, there are still some limitations in our work. First, our method focuses on explicit memory using RAG pipelines, and we primarily utilize CoT as the reasoning structure of agents. We will study implicit memory and other reasoning structures in future work. Additionally, the questions in HotpotQA might pose a leakage risk in the pre-training corpus for LLMs, and we will try to construct better evaluation datasets for this task.

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

# A PRE-TRAINED METRIC FUNCTIONS

## A.1 EMOTION SCORING FUNCTION

In addition to considering the semantic similarity between current states and memories, we propose incorporating emotional similarity as a factor for calculating matching scores. For each message, we represent its emotional content with eight dimensions: joy, acceptance, fear, surprise, sadness, disgust, anger, and anticipation (Huang et al., 2024b). This allows us to extract the emotion $\mathbf{h}_e(\phi_e; x)$ from message $x$. While most previous studies use LLMs for emotion scoring, this approach is hampered by randomness and a lack of comparative analysis of emotions across different messages. It also suffers increased computational expense due to frequent LLM inferences. To mitigate these challenges, we propose a pre-trained emotion scoring function leveraging contrastive learning. Specifically, we have

$$\mathbf{h}_e(\phi_e; x) = W_2^e \cdot \tanh(W_1^e \cdot \mathbf{h}_x^T + b_1^e) + b_2^e,$$

where $\mathbf{h}_x$ is the text embedding of $x$ and $\phi_e = \{W_1^e, W_2^e, b_1^e, b_2^e\}$ are trainable parameters. Then, the emotional similarity can be calculated as

$$d_{\text{emo}}(s^t, m_i) = \frac{\mathbf{h}_e(\phi_e; s^t) \cdot \mathbf{h}_e(\phi_e; m_i)^T}{||\mathbf{h}_e(\phi_e; s^t)|| \cdot ||\mathbf{h}_e(\phi_e; m_i)||}.$$

To optimize the emotion scoring function, we construct datasets for pre-training. The core assumption here is that the ability of LLMs to generate sentences with a specific sentiment is superior to their ability to discern the sentiment of given sentences. First, we generate a seed sentence $s_0$ without emotion. Next, we randomly select combinations $\{c_i\}_{i=1}^n$ of up to three emotions from the eight emotional dimensions. We then instruct the LLMs to generate sentences $s_i = \text{LLM}(s_0, c_i)$ containing the specified emotions based on this seed sentence and each emotional combination. Finally, we compile a dataset $\mathcal{D}^{\text{emo}} = \{(s_i, c_i)\}_{i=1}^n$ consisting of sentences with different emotions along with their corresponding emotion labels. Finally, we compile a dataset $\mathcal{D}^{\text{emo}} = \{(s_i, c_i)\}_{i=1}^n$ consisting of sentences with varying emotions and their corresponding emotion labels. Subsequently, we optimize our emotion scoring function with

$$\phi_e^* = \arg\min_{\phi_e} \frac{1}{|\mathcal{D}^{\text{emo}}|} \sum_{(s_i, c_i) \in \mathcal{D}^{\text{emo}}} [\mathbf{h}_e(\phi_e; x); c_i]^2.$$

To verify the effectiveness of our proposed method, we conduct extensive experiments across different baselines, and we present more details and in **Appendix A.3**.

## A.2 IMPORTANCE SCORING FUNCTION

In a similar approach, we pre-train an importance scoring function to evaluate various messages. It is crucial to differentiate between importance and relevance. Relevance pertains to the degree of semantic similarity between messages and is symmetrical in nature. Conversely, importance refers to the significance of specific information in relation to the current state and is asymmetrical. Therefore, we propose $\mathbf{h}_p(\phi_p; s^t) = W_1^p \cdot \mathbf{h}_{s^t}^T + b_1^p$ and $\mathbf{h}_p(\phi_p; m_i) = W_2^p \cdot \mathbf{h}_{m_i}^T + b_2^p$, where $\mathbf{h}_{s^t}, \mathbf{h}_{m_i}$ are the text embedding of $s^t, m_i$, and $\phi_p = \{W_1^p, W_2^p, b_1^p, b_2^p\}$. Then, we calculate the importance score with

$$d_{\text{imp}}(s^t, m_i) = \frac{\mathbf{h}_p(\phi_p; s^t) \cdot \mathbf{h}_p(\phi_p; m_i)^T}{||\mathbf{h}_p(\phi_p; s^t)|| \cdot ||\mathbf{h}_p(\phi_p; m_i)||}.$$

To optimize the importance scoring function, we construct datasets for pre-training. Initially, we select a query $q$ and a seed sentence $s_0$ containing minimal information. Subsequently, we incrementally enrich this seed sentence to generate new sentences $\{s_i\}_{i=1}^n$, thereby forming a partially ordered set. After that, we sample sentences from the same partially ordered set, forming positive and negative examples $(q, s^+, s^-)$. Finally, we obtain the dataset $\mathcal{D}^{\text{imp}} = \{(q, s_j^+, s_j^-)\}_{j=1}^m$ for contrastive learning. We optimize our importance scoring function with

$$\phi_p^* = \arg\min_{\phi_p} \frac{1}{|\mathcal{D}^{\text{imp}}|} \sum_{(q, s^+, s^-) \in \mathcal{D}^{\text{imp}}} \log \sigma \left[ d_{\text{imp}}(q, s^+) - d_{\text{imp}}(q, s^-) \right] - \log \sigma \left[ d_{\text{imp}}(q, s^-) - d_{\text{imp}}(q, s^+) \right].$$

## A.3 COMPARISON EXPERIMENTS AMONG DIFFERENT SCORING METHODS

We conducted experiments to assess the effectiveness of pre-trained metric functions within the memory retrieval process. Specifically, we compute NDCG@5 for ranked messages based on im-

Table 4: Results of pre-trained metric functions on testing datasets. Bold values represent the best results, while underlined values represent the second-best results.

| Methods | Base Models | Importance Scorer | | Emotion Scorer | | Cost |
| --- | --- | --- | --- | --- | --- | --- |
| | | nDCG@5 ↑ | IFR | MSE ↓ | IFR | |
| Random | Random | 0.498 | N/A | 2.685 | N/A | Low |
| Zero-shot Prompt | GPT-4o | 0.648 | 0.000 | 0.999 | 0.000 | High |
| | GPT-4o-mini | 0.826 | 0.000 | 0.983 | 0.000 | High |
| | Qwen-2.5 | 0.774 | 0.000 | 0.902 | 0.000 | High |
| | Llama-3.1 | 0.629 | 0.026 | 0.713 | 0.001 | High |
| Few-shot Prompt | GPT-4o | 0.672 | 0.000 | 0.904 | 0.000 | High |
| | GPT-4o-mini | 0.814 | 0.000 | 0.993 | 0.000 | High |
| | Qwen-2.5 | 0.578 | 0.000 | 0.814 | 0.006 | High |
| | Llama-3.1 | 0.439 | 0.661 | 0.942 | 0.001 | High |
| Ours (Learning) | E5-base-v2 | 0.978 | N/A | 0.491 | N/A | Medium |

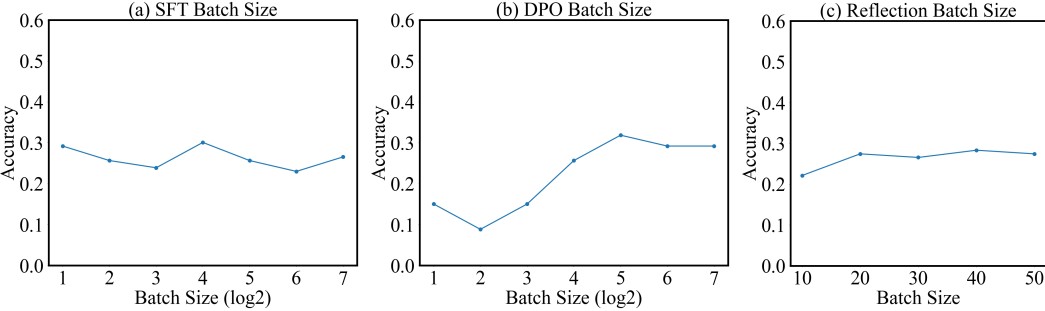

Figure 4: Results of different hyper-parameters.

portance scoring and MSE for emotional analysis across various dimensions. We also record the instruction failure rate (IFR) of prompting methods. For zero-shot and few-shot prompting techniques, we craft instructions for LLMs to generate scores within the range of $[0.0, 1.0]$ concerning importance and various emotional aspects of specific messages. Additionally, we employ GPT-4o, GPT-4o-mini, Qwen-2.5, and Llama-3.1 as the base models. For the random method, scores were independently generated from a uniform distribution between $0.0$ and $1.0$. We utilize E5-base-v2 as the base model for sentence embedding within our pre-trained metric functions. The results demonstrate that our pre-trained metric functions achieve notable improvements over zero-shot and few-shot prompting methods in predicting importance scores and emotional analysis. However, some open-source models exhibited instruction failures during scoring, contributing to instability in retrieval metrics.

## B   MORE EXPERIMENT DETAILS ON MEMDAILY

We conduct further experiments on the MemDaily dataset, focusing specifically on aggregative question-answering (QA) tasks. These tasks are the most challenging type, as they necessitate extended reasoning by recalling previous user messages. In line with our experiments on HotpotQA, we employ ReAct as the reasoning framework for our agents. To emulate an interactive scenario between users and agents, we divide all user messages in a specific trajectory into $k$ sequential blocks. These blocks serve as $k$ observations provided by the environment (*i.e.,* the user) to the agent. The interactions between the agent and the environment consist of $k + 1$ steps. During the first $k$ steps, we treat each $t$-th information block as the observation from the environment at step $t$, where the agent is not required to return any action. In the final step, we present a question to the agent as the observation and require it to return a predicted answer. We then compare the agent's final answer with the ground truth to calculate the Success Rate (SR). We set $k = 5$ in our experiments.

It should be noted that since the agent does not provide actions during the first $k$ steps, our proposed memory framework performs only one recall per trajectory. Consequently, there is only a single memory entity under this setting, and we optimize memory storage exclusively in off-policy and

Table 5: Overall performance across different baselines MemDaily.

| Inference | FUMemory | LTMemory | STMemory | GAMemory | MBMemory | SCMemory | MTMemory | Ours-on |
|---|---|---|---|---|---|---|---|---|
| Accuracy | 0.439 | 0.5366 | 0.5122 | 0.5366 | 0.4878 | 0.3171 | 0.2927 | **0.561** |

on-policy optimization. We employed Qwen-2.5 as the inference model for these tasks. The results are presented in **Table 5**. The experimental results indicate that our proposed method outperforms other baselines. Moreover, both GAMemory and LTMemory also demonstrate strong performance.

## C    MORE EXPERIMENT DETAILS OF HYPER-PARAMETER INFLUENCE

We further explore the influence of some significant hyper-parameters in our framework, including SFT batch size, DPO batch size, and reflection batch size. According to the results, we find that the best choice of SFT batch size is around 16, while the best DPO batch size is roughly 32. Additionally, we find that the accuracy is more sensitive to variations in DPO batch size, significantly diminishing when the values are particularly low. In contrast, the reflection batch size has a minor impact on performance, with accuracy remaining similar when it ranges from 20 to 50 samples per reflection.

## D   REPRODUCTION DETAILS

### D.1   ENVIRONMENT SETTINGS

We construct our environment based on HotpotQA, which includes hard, medium, and easy levels of difficulty. For each trajectory, the agent is required to give the correct answer to a question within a certain steps. For each step, the agent can observe feedback from the environment, and take the next action after that. There are two valid actions from agents: (1) `Search[entity]` means search the provided entity in Wikipedia. (2) `Finish[answer]` means give the final answer to the question.

We adopt the *fullwiki* mode in HotpotQA to make sure an interactive environment. In order to make the experiments more reproducible, we download the dumps file of Wikipedia. We obtain `wikipedia_en_all_nopic_2024-06.zim` (53.2GB) from Wikimedia Downloads [1], and implement a Wikipedia searcher with `libzim` package based on previous works. If the environment receives `Search[entity]`, it will search the given `entity` and return the full document of its Wikipedia page. If the environment receives `Finish[answer]`, it will compare `answer` with the ground truth and terminate this trajectory. If the environment receives other actions, it will return that the action is invalid. In our experiments, the maximum step is set as 5. The numbers of questions of hard, medium, and easy levels are 113, 109, and 107, respectively.

### D.2   AGENT SETTINGS

To better evaluate the memory capability of LLM-based agents, we standardize their reasoning structures as ReAct. For each step, the agent first receives the current state and executes the memory storage procedure. Then, it will execute the memory recall procedure to obtain a memory context. After that, it will think explicitly via LLM inference, and make the decision of actions based on the thought. Finally, it stores the thought and actions into memory and responds to the actions. The prompt of thinking and making action decisions is shown as follows.

---

The prompt of thinking of LLM-based agents.

You are a knowledgeable expert, and you are answering a question. You are allowed to search in Wikipedia to get information.
The question is: {question}. Now, you can choose to answer the question or search an entity on Wikipedia. Please think step by step to analyze how to choose the next action, and output it into one paragraph in concise. In previous steps, you have already accumulated some knowledge in your memory as follows: {memory_context}.

---

[1] https://dumps.wikimedia.org/kiwix/zim/wikipedia

> **The prompt of making the action decision of LLM-based agents.**
>
> You are a knowledgeable expert, and you are answering a question. You are allowed to search in Wikipedia to get information.
> The question is: {question}. You have thought step by step to analyze how to choose the next action as follows: {thought}.
> Now, you can choose to answer the question or search an entry on Wikipedia: (1) Search[entity], which searches the entity on Wikipedia and returns the paragraphs if they exist. (2) Finish[answer], which returns the answer and finishes the task. Your answer should be in concise with several words, NOT a sentence. Please generate the next action accordingly.
> Your output must follow one of the following two formats:
> Search[entity]
> Finish[answer]
> Here are some examples:
> Search[Alan Turing]
> Finish[no]
> Finish[Shanghai]
> In previous steps, you have already accumulated some knowledge in your memory as follows: {memory_context}

For the inference LLMs in our experiments, we utilize Qwen2.5-7B-Instruct, Llama3.1-8B-Instruct, and GPT-4o-mini.

### D.3 BASELINE SETTINGS

We implement baselines of memory methods based on MemEngine. For all the LLM inference inside memory methods, we utilize Qwen2.5-7B-Instruct as the backbone. For all the text embedding process, we adopt e5-v2-base model with 768 dimensions. For all the top-$k$ retrieval process, we set $k$ as 10 with cosine similarity. We set the maximum length of memory context as 8096 words. For all the summarization process, the prompt is as follows.

> **The prompt of summarization process.**
>
> Content: {content}
> Summarize the above content concisely, extracting the main themes and key information. Please output your summary directly in a single line, and do not output any other messages.

For GAMemory, we set the question number as 2, the insight number as 2, the reflection threshold as 0.3, the reflection top-$k$ as 2, and the prompt of reflection as follows.

> **The prompt of summarization process (generate question).**
>
> Information: {information} Given only the information above, what are {question_number} most salient highlevel questions we can answer about the subjects in the statements? Please output each question in a single line, and do not output any other messages.

> **The prompt of summarization process (generate insight).**
>
> Statements: {statements}
> What {insight_number} high-level insights can you infer from the above statements? Please output each insight in a single line (without index), and do not output any other messages.

For MBMemory, we set the forgetting coefficient as 5.0. For our method, the prompts of storage and utilization are shown as follows.

> **The prompt of storage process.**
>
> Observation: {observation}
> Hint: {hint}
> From the above observation and according to the hint, please extract critical informative points and summarize them into a concise paragraph. You should just output the result of summarization, without any other messages.

> **The prompt of utilization process.**
>
> Observation: {observation}
> Existing Memory: {memory_context}
> New Memory: {new_memory}
> Please merge the above new memory into the existing memory, which is useful to response the observation. You should remove the duplicated information to make it concise, but do not lose any information. You should just output the final memory after merge, without any other information.

For the hyper-parameters of off-policy training, we set the SFT learning rate as 0.0001, the SFT batch size as 16, the DPO learning rate as 0.0001, the DPO batch size as 16, and the reflection size as 40. For the hyper-parameters of on-policy training, we set the SFT learning rate as 0.0005, the SFT batch size as 16, the DPO learning rate as 0.0001, the DPO batch size as 16, the reflection size as 15, the sample batch size as 30, and the training epoch as 5.

