# OpenReview forum: "Learn to Memorize: Optimizing LLM-based Agents with Adaptive Memory Framework"
_ICLR.cc/2026/Conference — ICLR 2026 Conference Withdrawn Submission_

### Official Review · Reviewer_mHev · 2025-10-16

**Soundness:** 3
**Presentation:** 1
**Contribution:** 2
**Rating:** 4
**Confidence:** 3

**Summary:**

This paper presents an adaptive memory framework with a learnable memory cycle to support the decision-making process of LLM-based agents. By modeling the continuous interactions between agents and environments as an MDP, the authors define memory storage, retrieval, and utilization as parameterized policies to optimize. For memory retrieval, they design a MoE gate function to activate different metrics. For memory utilization, they use training datasets to optimize LLMs. For memory storage, they employ SFT and DPO to optimize the parameters of LLMs. Finally, after off-policy optimization of model parameters, they further conduct the on-policy optimization with online learning to avoid distribution shifts. Experiments and ablation studies on three datasets with various difficulty levels demonstrate that this adaptive memory framework helps better agent-based decision-making with different LLMs.

**Strengths:**

1.	The motivation of learning to memorize is a promising direction for LLM-based decision making.
2.	The proposed memory cycle effect is a good concept that the memory storage, retrieval, and utilization procedures influence each other.
3.	The authors test their framework with different tasks and LLMs. The ablation study also demonstrates the effect of each part.

**Weaknesses:**

1.	This whole framework is quite complex while the benefits of learnable memory are not significant enough for GPT-4o-mini and Qwen-2.5.
2.	Many details such as design choices are missing in the context. For example, why do the authors use the Bernoulli distribution for the stop signal in memory utilization?
3.	It seems that the training of this adaptive framework is costly, but no details are given.

**Questions:**

1.	How many metric functions do you use for each task? How do these metrics influence the performance of your method?
2.	Could the authors compare the computation efficiency of each method?
3.	How are the training datasets in memory utilization optimization constructed?
4.	It seems that the LLM should also be optimized. Is there any risk of the LLMs memorizing the answers to the questions?
5.	Why does your method have quite different performance on GPT-4o-mini or Llama-3.1?

---

### Official Review · Reviewer_2UtV · 2025-10-25

**Soundness:** 2
**Presentation:** 2
**Contribution:** 2
**Rating:** 2
**Confidence:** 4

**Summary:**

This paper proposes to build, retrieve from, and utilize memory for question answering tasks. Experiment results show certain improvements with open-source models.

**Strengths:**

**1. Complex Method Design with Concrete Problem Formulation.**
> The paper introduces all main components in the pipeline and presents mathematical formulations for them.

**Weaknesses:**

**1. Lack of Novelty in the Proposed Method.**
> Unlike what’s stated in the paper, there have been many works that consider the “memory cycle effect” by inducing, verifying, retrieving, and using memory entries [1,2]. Beyond covering non-paramatric update approaches as what’s being proposed in this work [3], some works also explore parametric updates [4]. There are many memory-related works besides what’s referenced in this comment. That being said, it is unclear what is the unique method this work proposes.

**2. Unclear Empirical Improvement.**
> From the results in Table 1, the proposed method only improves with open-source qwen and llama models (also unclear on the size), but underperforms existing methods with stronger GPT models. Therefore, it is unclear if the proposed method is effective and scales to future developments, especially as the backbone LMs keep getting stronger.

**3. Limited Coverage in Benchmarks.**
> This paper only experiments with one dataset, HotpotQA, which may not sufficiently cover the field of question-answering tasks. Yet, this paper attempts to claim its findings on “agents”, which is even less covered by this one dataset. Expanding the experiments to greater numbers and more agentic benchmarks would be helpful to address this concern.

[1] Wang, Zora Zhiruo, et al. "Agent workflow memory."

[2] Packer, Charles, et al. "MemGPT: Towards LLMs as Operating Systems." (2023).

[3] Chhikara, Prateek, et al. "Mem0: Building production-ready ai agents with scalable long-term memory."

[4] Wang, Yu, et al. "Memoryllm: Towards self-updatable large language models."

**Questions:**

N/A

---

### Official Review · Reviewer_QCNj · 2025-11-01

**Soundness:** 3
**Presentation:** 2
**Contribution:** 2
**Rating:** 4
**Confidence:** 4

**Summary:**

The paper learns a full memory policy for LLM agents across the retrieve,  utilize and store cycle: retrieval uses a MoE gate to weight relevance, utilization learns how to aggregate memories with a learned stopping rule, and storage uses task-specific reflection. The whole loop is trained with off-policy objectives plus an on-policy phase to reduce distribution shift.

**Strengths:**

(1) A learned MoE gate weights relevance/recency/importance/emotion per query instead of fixed cosine-only ranking.

(2) Storage uses learned reflection to write task-specific memories, improving future retrieval precision.

(3) An on-policy phase realigns retrieval/usage/storage with the agent’s own trajectories, reducing distribution shift and stabilizing the full loop.

**Weaknesses:**

1. In terms of the novelty, feeling more like a solid systems integration than a conceptual leap.

2. Main tables lack significance test, hard to judge variance, especially on easier splits.

3. Results are strongest on HotpotQA. A compact test on a different agent task (e.g., web-acting, tool use) would help generality.

4. Off-policy data construction and the supervision sources for SFT/DPO could be described more precisely to assess bias.

5. This paper suggests naive combinations can degrade before on-policy tuning. I think more concrete guidance to stabilize joint training would be needed.

**Questions:**

1. Can you add significance tests for the main metrics and the average steps/trajectory?

2. What’s the exact policy for off-policy data collection and how is it mixed with on-policy during training?

3. Do you have visualizations of MoE gate weights over states (when recency vs. importance vs. emotion dominates)?

4. Could you share a stability way that avoids degradation when combining procedures without full on-policy?

---

### Official Review · Reviewer_QuZ9 · 2025-11-01

**Soundness:** 2
**Presentation:** 3
**Contribution:** 2
**Rating:** 2
**Confidence:** 4

**Summary:**

The paper proposes an adaptive approach that enables an LLM-based agent to learn what should be remembered when solving a task.

**Strengths:**

1. The paper is relatively easy to read.
2. It addresses an important problem - memory mechanism in LLM-based agents.
3. A large number of diverse baselines are used in the experiments.

**Weaknesses:**

1. The Related Works section, especially the part on reinforcement learning, is too general. The purpose of such sections is to position the presented work relative to the most relevant existing studies, highlighting its novelty and significance. General information should instead be placed in the Background section.
2. The presentation of results in Table 1, split into two blocks for each dataset, is not convenient for readability. It would be better to display all results in a single block.
3. It is not entirely clear how the trajectories for training were collected and how high their quality is.
4. Only non-trainable methods are used as baselines, which makes it seem inappropriate to compare a trainable method against them.
5. It is questionable whether the formulation "the memory cycle effect" can be considered a contribution of the work. Agents with memory have been widely studied in reinforcement learning [1, 2, 3], and this formulation seems limited to the case of an explicit memory bank, whereas memory can also be stored in the hidden states or tokens of the model [4, 5].
6. The first and third contributions essentially describe the same result and should be combined.
7. The equations in the text are not numbered

I am willing to revise my evaluation if the shortcomings of the work are addressed.

**References:**
1. Lampinen, Andrew, et al. "Towards mental time travel: a hierarchical memory for reinforcement learning agents." Advances in Neural Information Processing Systems 34 (2021): 28182-28195.
2. Ni, Tianwei, et al. "When do transformers shine in rl? decoupling memory from credit assignment." Advances in Neural Information Processing Systems 36 (2023): 50429-50452.
3. Cherepanov, Egor, et al. "Unraveling the Complexity of Memory in RL Agents: an Approach for Classification and Evaluation." arXiv preprint arXiv:2412.06531 (2024).
4. Morad, Steven, et al. "Reinforcement learning with fast and forgetful memory." Advances in Neural Information Processing Systems 36 (2023): 72008-72029.
5. Cherepanov, Egor, et al. "Recurrent action transformer with memory." arXiv preprint arXiv:2306.09459 (2023).

**Questions:**

1. How appropriate is it to compare the proposed approach with non-trainable methods?
2. How much computational resources are required to train the model on a single task?
3. If an agent is trained in one environment, how well do the results transfer to another environment? For example, one with a similar structure but different actions and objects?
4. How does the addition of the emotional relevance metric affect the results? How specific is this metric to the datasets used, and how can it be adapted to other tasks?

---

### Note · Authors · 2026-01-05

I have read and agree with the venue's withdrawal policy on behalf of myself and my co-authors.